# Application of the Concept of Land Degradation Neutrality for Remote Monitoring of Agricultural Sustainability of Irrigated Areas in Uzbekistan

**DOI:** 10.3390/s23146419

**Published:** 2023-07-14

**Authors:** German Kust, Olga Andreeva, Darya Shklyaeva

**Affiliations:** Institute of Geography, Russian Academy of Sciences, Moscow 119017, Russia; andreevala@yandex.ru (O.A.);

**Keywords:** land degradation neutrality, sustainable land management, land degradation, Uzbekistan

## Abstract

A scientific approach to the assessment of trends in land changes based on the novel concept of Land Degradation Neutrality (LDN) was applied to monitor the sustainability of irrigated farmlands in test areas in Uzbekistan (the Andijan, Namangan, Fergana, and Syrdarya regions). The tool “Trends.Earth”, which was recommended by the UN Convention to Combat Desertification and developed as a special plugin for the Quantum GIS platform, was used to describe the dynamics of land degradation in the period 2001–2020. This study demonstrates the results of monitoring land productivity dynamics that reflect the investments in irrigation improvement during the last 10–15 years. A comparison between changes in land productivity measured via Normalized Difference Vegetation Index and its average value for the entire observation period is more informative than comparison with the initial 5-year period. More details could be noted through application of the “moving average” calculation method. The described trends demonstrate that the use of sustainable land management practices in the last decade led to a decreasing proportion of degraded lands compared to the average figure for the period 2001–2020 (from 25–40% to 10–20%). This trend is confirmed by reviewing state statistics and indicates the success of national policies and approaches to adaptation. However, the dynamics of land productivity in the study areas is diverse and includes “dry” and “humid” extremes, depending on climate fluctuations. Despite the generally positive trends identified across regions, the high dynamics of degraded hotspots and improved lands within certain areas confirm the instability of ongoing changes.

## 1. Introduction

The novel concept of Land Degradation Neutrality (LDN) has been actively developed in recent years after the adoption of the LDN definition by the 12th Conference of Parties of the United Nations Convention to Combat Desertification (UNCCD). According to this definition, LDN is “a state whereby the amount and quality of land resources necessary to support ecosystem functions and services to enhance food security remain stable, or increase, within specified temporal and spatial scales and ecosystems” [1].

Achieving LDN has been included in the United Nations Sustainable Development Goals (UN SDG). Target 15.3 directly aims to “achieve a land degradation-neutral world”, and its global indicator 15.3.1 defines total land degradation as “Proportion of land that is degraded over total land area” [2]. Achieving LDN requires three concurrent actions: avoiding new degradation of land by maintaining existing healthy land;reducing existing degradation by adopting sustainable land management practices;accelerating efforts to restore degraded lands to a natural or more productive state [3].

The scientific description of the LDN concept, principles used to select LDN indicators and establish the baseline for monitoring and further assessment, matrix of indicators, principles of mapping, and other details are provided in [4,5].

To achieve LDN, the UNCCD recommends using the “Trends.Earth” (TE) calculating module, which was developed as a special plugin for the Quantum GIS software [6]. TE is a tool used in the assessment of the state of terrestrial ecosystems using the time series of the main LDN indicators (dynamics of land cover, dynamics of land productivity, and dynamics of soil–organic carbon storage), which are integrated into TE and based on remote sensing and ground data. The tool is used to perform the qualitative and quantitative assessment of the proportion of degraded land (SDG indicator 15.3.1) and may help to identify the “hot spots” of land degradation processes.

In recent years, a sufficient number of studies have appeared that test the TE, propose approaches to improve calculations, and obtain the necessary information to solve certain scientific and practical issues [7,8,9,10,11]. In our earlier publications [12,13], we demonstrated that achieving LDN can be used as an integral indicator for sustainable land management (SLM) within particular territories, although its use is limited by certain biophysical and economic conditions. This study develops these previously expressed ideas and aims to demonstrate the possibility of assessing the effectiveness and sustainability of land use and land management practices applied to irrigated agricultural lands in Uzbekistan using LDN indicators in the transition period from a central planning-based to a market economy.

## 2. Materials and Methods

### 2.1. Researching Area

The study used the selected test territories of the Republic of Uzbekistan. Being part of the USSR from 1924 to 1991, Uzbekistan, with its favorable bioclimatic potential, became a strong cotton base in the Soviet Union; the production of rice, other grain crops, and fodder was also widely developed. This development required a significant expansion of irrigated areas and construction of new irrigation systems. At the same time, in practice, cotton became a monoculture that consumed a huge amount of water in its production and resulted in land degradation, such as widespread soil salinization of lands adjacent to irrigated areas, soil erosion on slopes, and flooding and salinization of lands in the areas of collector and drainage water discharges. The world-famous catastrophe of the drying up of the Aral Sea and associated desertification of vast areas was largely connected to ill-conceived over-irrigation in the basins of the Amudarya and Syrdarya rivers that feed the sea [14]. Overgrazing, improper functioning of irrigation and drainage systems, deforestation, and soil erosion processes are the main direct causes of desertification, as named by [15,16,17].

Climate aridity and aridization caused by irrational land use practices are important factors involved in desertification in Central Asian countries [18]. Li et al. [19] pointed out that since the 1960s, a trend of increase in the average annual temperature in Central Asia has been observed. Annual precipitation and soil moisture correspondingly decreased over the past few decades, especially from 2001 to 2014 [20].

In total, 75% of the land suitable for irrigation in Uzbekistan is saline (8.0 out of 10.7 million hectares), of which a proportion of highly saline land occupies 19% [21]. However, since the 1990s, an improvement in the soil potential of earlier abandoned lands with low fertility has taken place [22]. In the same period, agricultural restructuring in the country began. National statistical data clearly indicate that from 2001 to 2018, the release of a significant area of irrigated lands from the raw cotton production allowed Uzbekistan to increase grain production almost 4-fold, vegetable production almost 2.5-fold, potato production 6-fold, and horticulture production 4-fold. The production of animal feed has also increased significantly. Simultaneously, the restoration of irrigation and collector drainage systems continues, occurring jointly with improvement in irrigation approaches, including soil- and water-saving technologies. Significant areas of land abandoned as a result of salinization and desertification in the past are being returned to farming. As a result, during the period 2008–2017, the improvement in water supply to more than 1.7 million hectares of irrigated areas was achieved, and 2.5 million hectares of farmland were restored. The further implementation of the National Concept for the Effective Use of Land and Water Resources in Agriculture [23], which continues the results of effective management of irrigated lands, is designed for the period 2020–2030 and closely integrated with the UN SDGs for the period up to 2030.

The objective of our study was to trace these positive trends using LDN indicators and demonstrate how modern LDN monitoring tools help to track the dynamics of land degradation processes and set tasks for managing agricultural lands within individual administrative regions. For this purpose, we selected four administrative units of Uzbekistan: the Andijan, Namangan, Fergana, and Syrdarya regions (Figure 1). This choice was not made randomly, since in these areas, arable land is almost entirely represented by irrigated farmland. Orchards, hayfields, homesteads, and even some forest lands are also irrigated. Compared to the territory of Uzbekistan in total, for which the proportion of irrigated land is on average about 10% of the territory, in selected regions, the share of irrigated land is more than 50% of all land, even exceeding the overall area of farmland (Table 1).

### 2.2. Methods 

The TE calculation module [8] was used as the main tool for the assessment of the LDN achievement and to monitor its indicators. Global databases, including those based on remote sensing data [9], were used as a dataset for TE (Table 2).

ESA CCI LC data sources are represented by the Meris (MEdium Resolution Imaging Spectrometer) satellite systems, with a spatial resolution of 300 m; the Vegetation system of the SPOT and PROBA-V satellites, with spatial resolutions of 1 km; and the NOAA satellite AVHRR sensor; with a spatial resolution of 1 km [25]. Spatial data in the form of maps produced annually from 1992 to 2020 introduce global land cover [26] in 22 or 36 land cover types [27]. For the dynamics of land productivity, materials from the Terra-MODIS (Moderate-resolution Imaging Spectroradiometer) system were used. MODIS data were collected every 16 days. The MOD13Q1 product made it possible to generate regular data on the Normalized Difference Vegetation Index (NDVI). SoilGrids, which is a tool that was used as a data source for the analysis of the dynamics of soil organic carbon (SOC), is a global database that contains information on the spatial distribution of soil properties and is based on machine learning methods for mapping [6].

Since the structure and composition of land cover in the studied test regions have hardly changed over the past 30–40 years, and the data of soil studies were not updated in the same period, it can be assumed that the dynamics of land degradation/improvement will mainly be reflected in changes in just one of the global LDN indicators—the dynamics of land productivity. This indicator contributes the main input used in the assessment of the state of land for studied regions using the TE tool.

NDVI, which is computed using remote sensing information from the red and near-infrared portions of the electromagnetic spectrum, is one of the most used surrogates of net primary productivity (NPP). To simplify the interpretation of results, bi-weekly products from MODIS and AVHRR were used to compute mean annual integrals of NDVI, which were then used to calculate three productivity sub-indicators derived from NDVI time series data: *trajectory*, *state* and *performance*. *Trajectory* measures the rate of change in primary productivity over time. TE computes a linear regression at the pixel level to identify areas that experience changes in primary productivity for the period under analysis. A Mann–Kendall non-parametric significance test was then applied, which considered as significant changes only those that showed a *p*-value ≤ 0.05. Positive significant trends in NDVI indicated potential improvement in land condition, and negative significant trends indicated potential degradation. The *Productivity State* sub-indicator allowed the detection of recent changes in primary productivity compared to a baseline period. For this purpose, the annual integrals of NDVI for the baseline period were used to compute a frequency distribution for each pixel. Next, the mean NDVI of the baseline period was compared to the mean NDVI for the comparison period. For this procedure, the percentile classification from 1 (lowest class) to 10 (highest class) was recommended: if the difference in class between the comparison and the baseline period was ≤2, the pixel was potentially degraded. If the difference was ≥2, the pixel indicated a recent improvement in terms of primary productivity. Pixels with small changes were considered stable. The *Productivity Performance* sub-indicator measured local productivity relative to other similar vegetation types in similar land cover types or bioclimatic regions throughout the study area. The computed mean NDVI in the time series for each pixel was compared to the mean NDVI for ecologically similar units (as a unique intersection of land cover and soil type). If the observed mean NDVI was lower than 50% of the maximum productivity, the pixel was considered potentially degraded for this indicator [6].

The three productivity sub-indicators, based on [6], were then combined, as indicated in Figure 2. For SDG 15.3.1 reporting, the 3-class indicator was required, though TE also produced a 5-class indicator, which took advantage of the information provided by State to inform the type of degradation occurring in the area.

## 3. Results

Although the main purpose of the TE tool was primarily to calculate the proportion of land degraded in a certain period (SDG indicator 15.3.1), some peculiarities of this tool allow us to monitor the dynamics of land degradation and its individual indicators for selected time periods. These facilities can be demonstrated using the following examples.

At the first stage of our work to assess changes and provide a retrospective monitoring [28] of the state of land, while considering the recommendations of the TE [6] for the selected regions, we built diagrams of the dynamics of productivity and land degradation from 2001 to 2020 with a five-year step (recommended by [6] as a minimum period for monitoring), while the initial five-year period 2001–2005 was taken as the baseline. As can be seen in the diagrams in Figure 3, the progress in land degradation dynamics with small deviations repeats the dynamics of land productivity. This result confirms our initial assumption that the two others global LDN indicators do not significantly affect the assessment, since the state of land cover and the stocks of soil organic carbon in studied areas do not change significantly. Indeed, the literature data confirm that the structure of land use in these areas remains almost unchanged for several decades, and the stocks of organic carbon are also relatively stable [29,30,31]. The soil humus content in irrigated soils mostly varies within relatively wide limits due to different soil textures, which ranges from 0.6–0.9% in the upper layer in light and newly irrigated soils to 1.5–1.8% in heavy texture and old irrigated soils.

Results obtained demonstrate that selecting the first five years as the baseline period is reflected on the graphs, the earliest point on which is characterized by the proportion of stable and improved territories being close to 100%, which creates a false illusion that only a deterioration in indicators is further noted. Moreover, conclusions on the use of similar starting periods for comparison can be found in [17,32,33,34]. These conclusions do not correspond to the state statistics data [35] obtained via traditional field-based methods, which indicate certain positive trends in the state of land, productivity growth, and improvement in irrigation facilities that resulted from present state policy and significant investments in land reclamation measures and increasing land fertility. We conceded that this uncertainty was a result of using the initial five years as a baseline period; therefore, no change over this period was reflected in the graph.

To test this assumption, we proposed a special method, in which average values of indicators collected over the entire observation period of 2001–2020 were used as the baseline (Figure 4). In this case, the values for each period were compared to their long-term averages, and, thus, the diagrams obtained reflected the deviation from this average. The results received, therefore, did not indicate the absolute values of the areas of degraded or improved land compared to the initial period, but reflected the relative (to the average) deterioration or improvement in indicators compared to the long-term average. This approach is closer to the essence of the LDN concept, which refers to the balance between degradation and improvement. Thus, in our opinion, it is possible to demonstrate a more adequate picture of the monitoring of productivity and land degradation for certain periods. From Figure 4, it can be seen that compared to the 20-year average, the first years of this period were relatively negative, while in recent years, there was a significant improvement in all areas studied.

To refine and improve the picture of the dynamics, we used an additional methodological technique, which consisted of graphs that were plotted based on the “moving average” principle [36]. Using a five-year assessment period within the same time period allows us to level possible acute climatic fluctuations (the first time that such an approach for assessing the dynamics of LDN was applied was in [37]).

The moving average method involves a transition from the initial values Xn of the time series to the average values Xav calculated on their basis, which were recorded over a certain time interval (n) [32]. This method helps us to demonstrate the main trends in the form of time series in a smoother form [38]. This approach is based on the fact that the variance in the calculated average values is n times lower than the variance in the initial values of the time series. In our study, the values of the indicators were calculated for the following periods: 2001–2005, 2003–2007, 2006–2010, 2008–2012, 2011–2015, 2013–2017, and 2016–2020. Xav values determined using the moving average method were calculated for Xn for 2003, 2005, 2008, 2010, 2013, 2015, and 2018, respectively. Consequently, the updated charts were built with seven points that averaged a 5-year time series instead of the four points originally used, which reflected the dynamics in more detail.

The diagrams obtained using this approach (Figure 4) demonstrate that the dynamics of land productivity and the related dynamics of the proportion of degraded lands vary in the selected pilot regions. If, in Andijan and Fergana, there is a clear trend of land improvement from the period 2006–2010 onwards, it is clear that in Namangan and Syrdarya, the dynamics are more complicated, and after the periods of deterioration, improvement is only first noted after the period 2013–2017. These results, apparently, are associated to a large extent with climatic droughts, which were especially serious in the period 2001–2014, as indicated by [16,34,35]. We associate the periods of lesser impact of climatic conditions in Andijan and Fergana with earlier investment in water and land management and irrigation in these regions than in Namangan and Syrdarya.

A more detailed picture of the dynamics of land productivity can be seen in the maps presented in Figure 5. The maps clearly demonstrate that despite the relatively positive trends in land state averaged over the regions (compared to data in Figure 4), only a few hotspots and areas of improvement remain fairly constant over time. Most of the problematic zones relocate, which indicates high land productivity in certain periods of time and confirms the unstable nature of land management in the studied regions.

## 4. Conclusions

The tool Trends.Earth, which was developed for the integral LDN assessment, can be applied to trace changes in the state of agricultural land in Uzbekistan. The dynamics of land productivity, in this respect, serves as the most informative indicator measured through the NDVI.

Time series of land productivity for the period 2001–2020, which were obtained using TE, differ depending on the selected baseline period. To assess the dynamics within relatively short periods of time (15–25 years), a comparison between changes in this LDN indicator and the average value for the entire observation period is more informative than a comparison with the initial period. More details of the dynamics of land state could be noted through the application of the “moving average” calculation method.

The example of studied areas characterized by a high proportion of agricultural land in relation to overall land cover, as well as a high proportion of irrigated land within them, demonstrates that the use of SLM practices in the last decade can lead, in general, to a decrease in the total proportion of degraded land compared to the average figure for the period 2001–2020. This trend is confirmed by state statistics and indicates the success of national and regional policies and approaches to help the country to adapt to the backdrop of unfavorable climatic changes in Central Asia.

However, the dynamics of land productivity in the study areas are diverse and characterized by extremes, such as relatively drier or more humid years, that are confirmed by analyzing data on climatic fluctuations. Despite the generally positive trends across the studied regions, the high dynamics of degraded hotspots and improved land within certain areas confirm the unstable nature of the ongoing changes, which require special attention from policy makers to protect agriculture and land management.

## Figures and Tables

**Figure 1 sensors-23-06419-f001:**
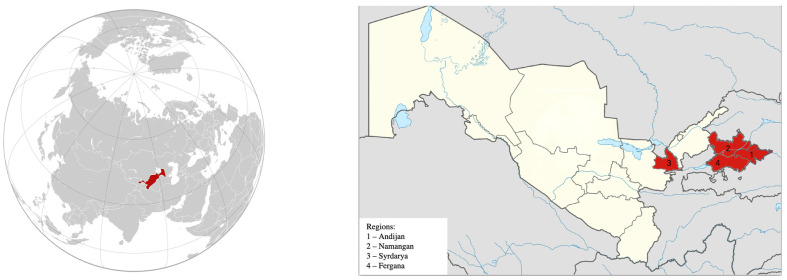
Pilot areas studied in Uzbekistan.

**Figure 2 sensors-23-06419-f002:**
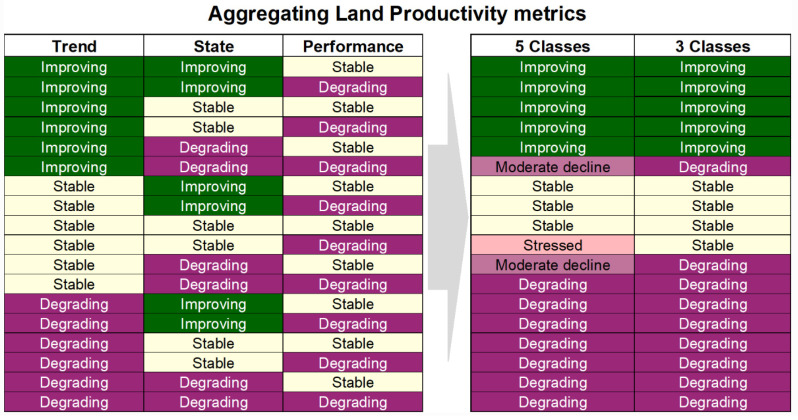
Use of combinations of land productivity sub-indicators derived from the NDVI calculations [6].

**Figure 3 sensors-23-06419-f003:**
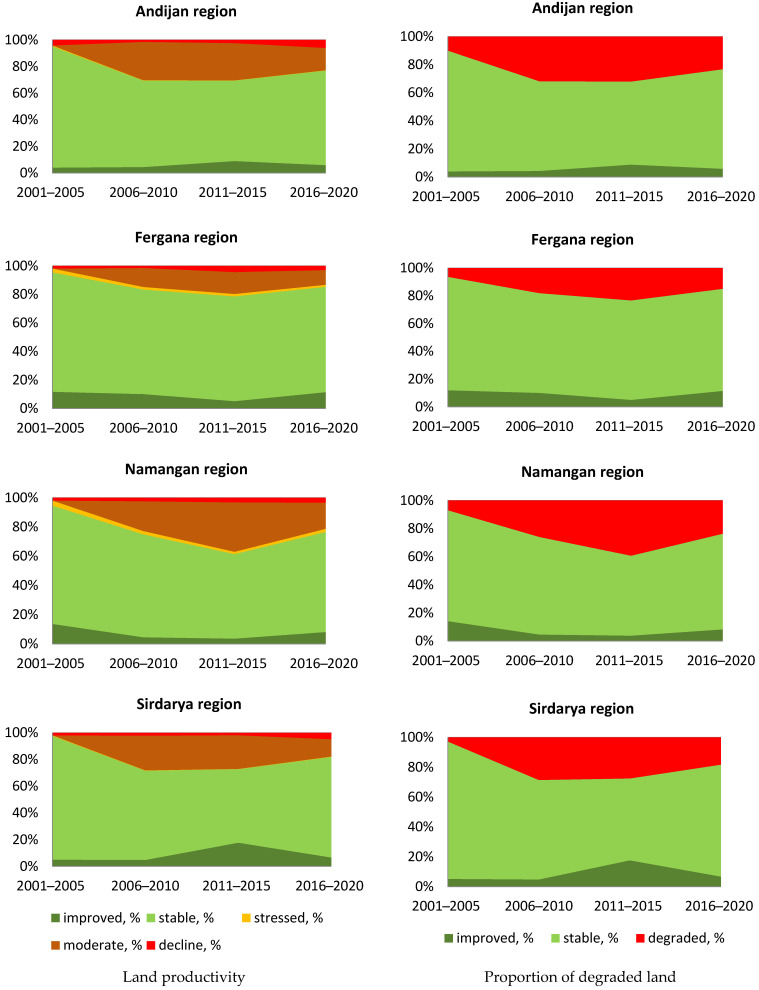
Dynamics of land productivity (**left column**) and proportion of degraded land (**right column**) compared to initial baseline period 2001–2005.

**Figure 4 sensors-23-06419-f004:**
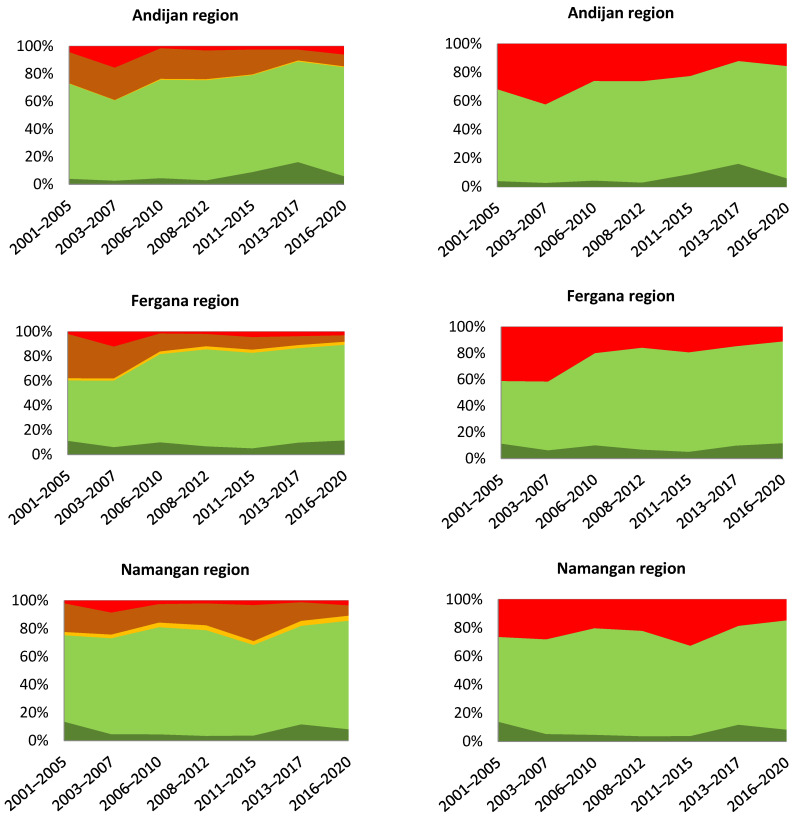
Dynamics of land productivity (**left column**) and proportion of degraded land (**right column**) compared to average baseline period 2001–2020 (see legend to Figure 2).

**Figure 5 sensors-23-06419-f005:**
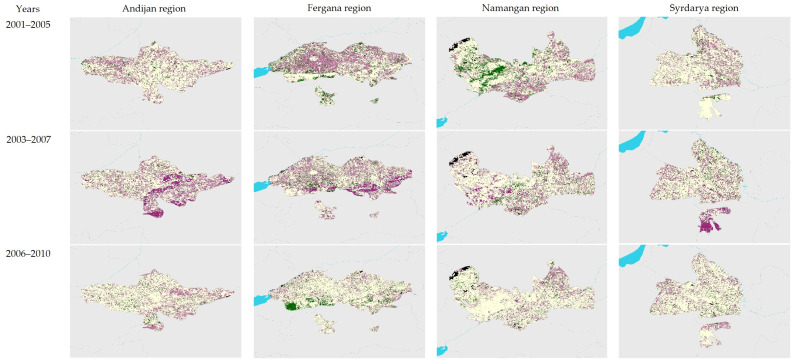
Recent dynamics of land productivity compared to average baseline for the period 2001–2020 (see legend in Figure 2).

**Table 1 sensors-23-06419-t001:** Area of agricultural and irrigated lands in pilot regions, as shown in [24].

Region	Total Area, Thousand Hectares	Agricultural Lands		Irrigated Lands in Total (Incl. Farmlands, Forests, Horticulture, etc.)	
		Thousand Hectares	%	Thousand Hectares	%
Andijan	430.3	255.4	59.4	274.2	63.7
Namangan	718.1	387.9	54.0	290.0	40.4
Fergana	700.5	320.4	45.7	368.5	52.6
Syrdarya	427.6	287.4	67.2	286.3	67.0
Uzbekistan (in total)	44,892.4	25,639.0	57.1	4329.0	9.6

**Table 2 sensors-23-06419-t002:** Datasets used for Trends.Earth calculations.

LDN Indicator	Data Set	Resolution (Meters)	Time Period
Land Cover	ESA CCI LC	300	1992–2020
Land Productivity (NDVI)	MOD13Q1	250	18 February 2000–17 February 2023
Soil Organic Carbon	Soil Grids	250	2020

## Data Availability

Some datasets used in the study can be can obtained from the following sources of the Republic of Uzbekistan: Presidential Decree on Measures for the Effective Use of Land and Water Resources in Agriculture. 2019. Available online: https://lex.uz/ru/docs/4378524#4380363 (accessed on 14 June 2023). (In Russian); Statistics Agency under the President of the Republic of Uzbekistan. Agriculture. Available online: https://stat.uz/en/official-statistics/agriculture (accessed on 14 June 2023).

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
