# Peer review of "Application of the Concept of Land Degradation Neutrality for Remote Monitoring of Agricultural Sustainability of Irrigated Areas in Uzbekistan"

_sensors, 2023, doi:10.3390/s23146419_

Round 1

Reviewer 1 Report

The manuscript Application of the Concept of Land Degradation Neutrality for Remote Monitoring of Agricultural Sustainability of Irrigated Areas in Uzbekistan”  used a novel scientific approach to evaluate the proportion of degraded lands basing on the concept of Land Degradation Neutrality (LDN) was applied to the assessment of trends in land changes in test areas of Uzbekistan (Andijan, Namangan, Ferghana and Syrdarya regions).

The manuscript is prepared professionally. It includes a well-crafted abstract and an exhaustive introduction that justifies the research undertaken. The introduction points to the deficiencies in the literature on the subject. The aim is clearly defined. Modern analytical methods were used in the research. The discussion of the results is well prepared. The conclusions are well-defined. The illustrative material is appropriate.

In my opinion, the manuscript after corrections, will be suitable for publication in a journal.

Detailed comments:

Abstract: Too long and must be included some numeric data from results.

Do not use abbreviations when use first time

Introduction - The introduction is enough in my opinion. Introduction needs some minor changes

Line 41....spatial scales and ecosystems [1]. Please use more references and I suggest below ones.

Fatemeh, S.; Dervishan, AK.; Golosov, V.; Zare, MR.; Spalevic, V. Influence of land use on changes of sediment budget components: western Iran case study. Turk. J. Agric. For. 2022,  46 (6):838-851. https://doi.org/10.55730/1300-011X.3046.

Metin, AE.; Caglak, S. Assessment of the effect of land use change on bioclimatic comfort conditions in Usak province. Turk. J. Agric. For. 2022, 46 (5):632-641. https://doi.org/10.55730/1300-011X.3032.

Line 84-85 please follow journal writing rules (Avetisyan, 2016). Li et al. 84 (2017) 

The manuscript Application of the Concept of Land Degradation Neutrality for Remote Monitoring of Agricultural Sustainability of Irrigated Areas in Uzbekistan”  used a novel scientific approach to evaluate the proportion of degraded lands basing on the concept of Land Degradation Neutrality (LDN) was applied to the assessment of trends in land changes in test areas of Uzbekistan (Andijan, Namangan, Ferghana and Syrdarya regions).

The manuscript is prepared professionally. It includes a well-crafted abstract and an exhaustive introduction that justifies the research undertaken. The introduction points to the deficiencies in the literature on the subject. The aim is clearly defined. Modern analytical methods were used in the research. The discussion of the results is well prepared. The conclusions are well-defined. The illustrative material is appropriate.

In my opinion, the manuscript after corrections, will be suitable for publication in a journal.

Detailed comments:

Abstract: Too long and must be included some numeric data from results.

Do not use abbreviations when use first time

Introduction - The introduction is enough in my opinion. Introduction needs some minor changes

Line 41....spatial scales and ecosystems [1]. Please use more references and I suggest below ones.

Fatemeh, S.; Dervishan, AK.; Golosov, V.; Zare, MR.; Spalevic, V. Influence of land use on changes of sediment budget components: western Iran case study. Turk. J. Agric. For. 2022,  46 (6):838-851. https://doi.org/10.55730/1300-011X.3046.

Metin, AE.; Caglak, S. Assessment of the effect of land use change on bioclimatic comfort conditions in Usak province. Turk. J. Agric. For. 2022, 46 (5):632-641. https://doi.org/10.55730/1300-011X.3032.

Line 84-85 please follow journal writing rules (Avetisyan, 2016). Li et al. 84 (2017) 

Author Response

Dear Reviewer 1,

We acknowledge your recommendations to improve the manuscript under review. Please, find our feedback in the attached file.

Reviewer 2 Report

This article attempts to assess Land Degradation Neutrality in Uzbekistan for irrigated agricultural lands using the Trends.Earth method. It explores the issue of selecting baseline values through the application of the "moving average" calculation method. This represents a commendable effort to address the challenge of baseline value selection in long time series analysis, offering important guidance for other scientific research endeavors. However, substantial revisions are still required in terms of the manuscript's presentation to meet the publication requirements. The focus should primarily be on elucidating the determination of Land Degradation using trajectory, state, and performance indicators, as well as clarifying the representation of the 5 Classes and 3 Classes. It is necessary to provide a detailed description of the meanings associated with the categories of improving, moderate decline, stable, stressed, and degrading, along with a comprehensive explanation of the criteria used for their classification. Additionally, the values of the indicators were calculated for the following periods: 2001-2005, 2003-2007, 2006-2010, 2008-2012, 2011-2015, 2013-2017, and 2016-2020. Specifically, Xav values were calculated for Xn in the years 2003, 2005, 2008, 2010, 2013, 2015, and 2018 using the moving average method. The criteria for selection should be described in detail. In conclusion, as a guiding manuscript, it is crucial to enhance the comprehensive description of the methodology.

Author Response

Dear Reviewer 2,

Thank you for your comments and recommendations. You can find our feedback in the attached file.

Regards 

Reviewer 3 Report

In this manuscript, the authors discuss indexes -including some derived from spaceborne Earth observation- that can be used for estimating land degradation in Central Asian countries, and their combinations.

I regret I cannot recommend this manuscript for publication. I found it difficult to understand what is the actual scientific contribution proposed by the authors; the discussion does not higlight it clearly, but rather presents a series of existing indexes and remarks on how they may be used and combined. 

The evaluation of results, consisting of time trends for degradation classes from the proposed combinations, is qualitative.

The main issues with the manuscript are, in my opinion:

- lack of visible, relevant scientific novelty;

- the topic discussed, i.e. mapping of land degradation in Central Asia, fits the scope of the Sensors journal quite weakly; Sensors focuses on sensing devices, the data they produce, and how to process these latter. The submitted manuscript is confined to the use of indexes derived from remotely sensed data, and the process to obtain such indexes is taken as a fact. Remote sensing as such is out of the picture. I see the paper more suitable for a journal in the domain either of environmental studies, or GIS.

- at some points, poor English (e.g. page 2, row 55 "a possibility for qualitatively and quantitatively assessment..."; row 184, page 5"the data of the first five-year period are also belong to the period..."; there are more)

I suggest the authors to rework the manuscript to highlight its main points in terms of advancement with respect to the state of the art in land degradation mapping, and resubmit it to a journal in the domain of environmental studies.

Please have the manuscript cross-checked by a qualified English speaker.

Author Response

Dear Reviewer 3,

Thank you for your comments. We think  that your negative conclusion is mainly related to the fact that you probably did not consider that the manuscript was submitted for the Special Issue of SENSORS "Remote Sensing for Water Monitoring in Agricultural Management and Development". Some more details you can find in the attached file

Regards 

Round 2

Reviewer 3 Report

The text is now more fluent and more informative, but even considering it was submitted to a Special Issue, I think it fits the scope only partly. It may be of interest to environmental degradation studies, but not to an audience interested in sensors and remote sensing.

I found it difficult to assess the real scientific advancement, and this is not connected with LDN per se, as per the authors' reply, although I found the current additions to be of help. LDN is simply the tool used, but the "methods" subchapter describes pre-existing algorithms and methods from [6], including figure 2. More details are provided in the "results" chapter, where however the main novelty seems to consist of the application of a moving average step, and the rest is mostly interpretation of existing information. The main point there is linked to the environmental conclusions that may be drawn from using known indexes.

I sill doubt the proposed manuscript fits the interest of the readers and its contents align with the expected level of scientific advancement. I will however let the Editor decide.